# NATLM: DETECTING DEFECTS IN NFT SMART CONTRACTS LEVERAGING LLM

## ABSTRACT

Security issues are becoming increasingly significant with the rapid evolution of Non-fungible Tokens (NFTs). The potential defects in NFT smart contracts could lead to substantial financial losses if exploited. To tackle this issue, this paper presents a framework called NATLM (**NFT A**ssistan**t LLM**), to detect potential defects in NFT smart contracts. NATLM effectively identifies 4 common types of vulnerabilities in NFT smart contracts, including ERC-721 Reentrancy, Public Burn, Risky Mutable Proxy, and Unlimited Minting. Relying exclusively on large language models (LLMs) for defect detection can lead to a high false-positive rate. To improve it, NATLM integrates static analysis with LLMs, specifically Gemini Pro 1.5. Initially, NATLM employs static analysis to extract structural, syntactic, and execution flow information from the code, represented through Abstract Syntax Trees (AST) and Control Flow Graphs (CFG). These extracted features are then combined with vectors of known defect examples to create a matrix for input into the knowledge base. Subsequently, the feature vectors and code vectors of the analyzed contract are compared with the contents in the knowledge base. Finally, the deep semantic analysis capabilities of LLM are used to identify defects in NFTs. Experimental results indicate that NATLM analyzed 8,672 collected NFT smart contracts, achieving an F1 score of 88.94%, outperforming other baselines.

## 1 INTRODUCTION

Ethereum Buterin et al. (2014) introduced the paradigm of Turing-complete smart contracts Zheng et al. (2017), which are self-executing agreements composed of code stored on the blockchain. The Ethereum Improvement Proposal EIP-721 William et al. (2018) introduced the ERC-721 token standard, enabling developers to create unique digital assets. Non-fungible tokens (NFTs) Fairfield (2022) are non-replicable digital assets or unique identifiers managed on the blockchain, used to allocate, link, or prove ownership of distinct physical and digital goods. However, as the NFT market rapidly grows, the security issues surrounding smart contracts are gradually coming to light, becoming a key factor hindering the market's healthy development. Once deployed, smart contracts are immutable. While this immutability ensures transparency and reliability in transactions, it also means that attackers can exploit these flaws for malicious purposes if vulnerabilities exist, potentially leading to significant financial losses. These vulnerabilities mainly stem from logical flaws, contract design defects, and failure to consider potential security risks during development adequately.

Traditional methods for detecting vulnerabilities in smart contracts, such as static analysis tools like SlitherFeist et al. (2019) and TruffleHartel & van Staalduinen (2019), offer comprehensive reviews of smart contracts without executing the code, identifying common security flaws. However, these methods have certain limitations. Static analysis often relies on predefined rules and patterns, making it prone to false positives and false negatives when handling complex smart contracts, thus complicating the accurate identification of real security threats. The emergence of large language models (LLMs) Wei et al. (2022) has sparked widespread interest and research. LLMs, such as CodeBERT Feng et al. (2020) and Gemini Google AI Blog (2024), are models based on the Transformer Devlin et al. (2018) architecture, possessing powerful natural language processing capabilities that enable them to learn complex semantics and contextual relationships from vast amounts of textual data. These models can not only understand and generate human-like natural language but can also be applied to code understanding and vulnerability detection. By learning from a large corpus of smart contract code and security audit reports, LLMs can effectively capture semantic features within the

code, identify potential security risks, and generate detailed explanations of vulnerabilities. This capability gives LLMs a unique advantage in vulnerability detection, and they have already been employed in smart contract vulnerability detection Boi et al. (2024) and software vulnerability detection Purba et al. (2023) Saji Mathews et al. (2024). However, LLMs primarily analyze code based on textual semantics but face limitations in precisely parsing code hierarchy, variable dependencies, and control structures. Additionally, LLMs may produce misleading results or interpretations, especially when dealing with complex or ambiguous code snippets, increasing the risk of false positives or negatives. Current research finds David et al. (2023) that while LLMs can accurately identify some vulnerabilities in smart contract security audits, their high false positive rate still requires the involvement of manual auditors.

This study proposes a tool called NATLM to explore how combining large language models (LLMs) and static analysis methods can detect defects in NFT smart contracts. Specifically, we focus on four common NFT smart contract defects: ERC-721 Reentrancy, Risky Mutable Proxy, Public Burn, and Unlimited Minting defects Yang et al. (2023). The main contributions of this paper are as follows:

- To the best of our knowledge, this work is the first to propose NATLM, a novel framework that combines static analysis with Gemini Pro 1.5 for detecting defects in NFT smart contracts. NATLM employs static analysis to extract critical features, including variables, functions, and control flow structures, directly from the contract's codebase. These features constitute a comprehensive knowledge base, which enhances both the accuracy and the efficiency of defect detection.

- NATLM employs a dual-stage detection strategy that combines feature fusion and advanced LLM analysis. The feature vectors and code vectors are fused to match similar data in the Knowledge Base, and then the Gemini model is utilized to identify defects.

- We conduct extensive experiments to evaluate the effectiveness of NATLM across multiple defects in NFT smart contracts. The overall precision is calculated to be 87.72%.

## 2 THE NATLM FRAMEWORK

In this section, we present the overall design of NATLM. The overall framework is illustrated in Figure 1. The NATLM aims to detect defects in NFT smart contracts by combining static analysis with LLM. The overall workflow of the tool consists of three main phases, including Feature & Knowledge Base Extraction, Knowledge Base Retrieval, and LLM Reasoning.

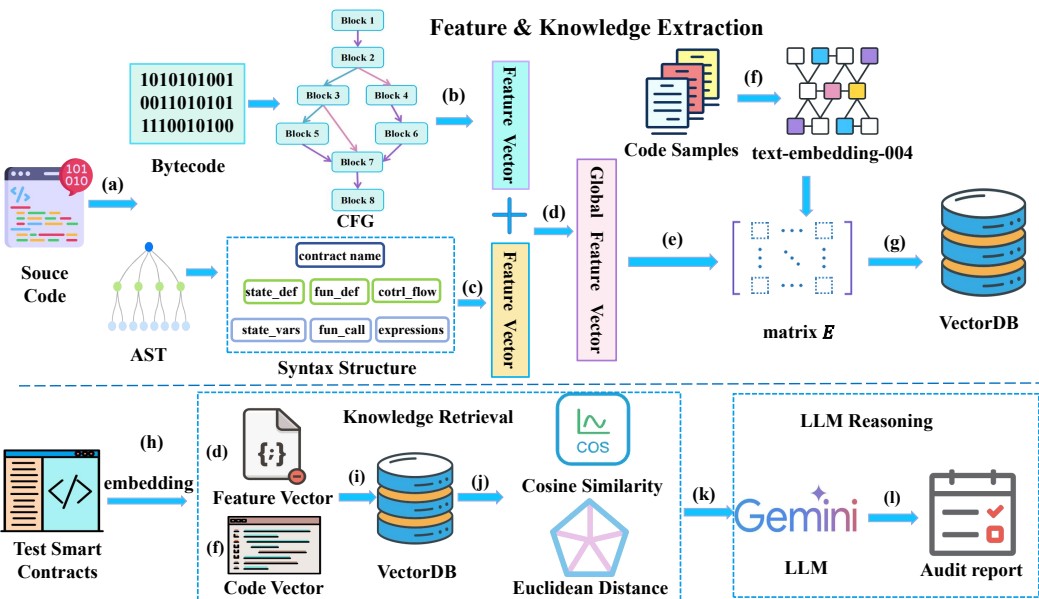

Figure 1: The Overall Framework of NATLM

## 2.1 Feature & Knowledge Base Extraction

During the feature extraction and knowledge base construction phase, NATLM first parses the NFT smart contract source code to generate an Abstract Syntax Tree (AST), which reveals the structure and syntactical hierarchy of the code. We use the Solidity compiler (solc) to parse the source code into an AST, mapping the structure of the code (such as function definitions, variable declarations, and control structures) into a tree format. The purpose of generating the AST is to capture the syntactic and semantic information within the contract, extracting key elements such as state variables, function definitions, and function call records while documenting the locations of these calls.

Once the AST is generated, we use CodeBERTRen et al. (2023) to extract feature vectors from the AST. CodeBERT is a pre-trained language model designed for code understanding, capable of capturing both syntactic and semantic information in the code. Before feeding the AST into CodeBERT, we first linearize the AST by performing a depth-first search (DFS) on the tree structure, flattening each node into a sequential order. The linearized sequence of nodes is then passed into CodeBERT's tokenizer, which splits complex identifiers into subtokens (e.g., 'mintTokens' becomes '[mint, Tokens]') to handle compound or rare terms. Special tokens '[CLS]' and '[SEP]' are added at the beginning and end of the sequence, respectively, to mark the sequence boundaries. Additionally, position embeddings are applied to each token to retain their relative order in the sequence. The tokenized sequence, consisting of word embeddings and positional encodings, is passed through the embedding layer of CodeBERT. Each subtoken is mapped to a fixed-size vector $(\mathbf{e}_i \in \mathbb{R}^d)((d = 768))$, to reduce computation overhead, a linear projection layer at Equation 1 is applied.

$$\mathbf{e}'_i = W_{\text{proj}} \cdot \mathbf{e}_i + b_{\text{proj}} \tag{1}$$

where $W_{\text{proj}} \in \mathbb{R}^{256 \times 768}$ is the projection matrix, and $b_{\text{proj}}$ is the bias term. The resulting vector $\mathbf{e}'_i \in \mathbb{R}^{256}$ serves as the final input representation for each subtoken.

These input embeddings are passed through the transformer encoder layers of CodeBERT, where each layer applies a multi-head self-attention to compute the relationships between different tokens in the sequence. L2 regularization is applied to all learnable parameters to prevent overfitting during fine-tuning. Additionally, a dropout layer with a rate of 0.1 is applied after the feed-forward neural network (FFN) layers to prevent overfitting further. The FFN computation is given by Equation 2.

$$\text{FFN}(x) = \text{ReLU}(W_1 x + b_1)W_2 + b_2 \tag{2}$$

where $W_1$ and $W_2$ are weight matrices, and $b_1$ and $b_2$ are bias vectors. After passing through multiple encoder layers, CodeBERT produces an embedding matrix $\mathbf{H}$ for the entire sequence of nodes: $\mathbf{H} = [\mathbf{h}_1, \mathbf{h}_2, \dots, \mathbf{h}_N]$ where $\mathbf{h}_i \in \mathbb{R}^d$ represents the feature vector for the $i$-th AST node, and $N$ is the number of nodes in the AST. To obtain a global feature vector representing the entire AST, a mean pooling operation as follows is applied over all node-level embeddings.

$$\mathbf{x}_{\text{ast}} = \frac{1}{N} \sum_{i=1}^{N} \mathbf{h}_i$$

This pooling operation aggregates the individual node embeddings into a single vector $\mathbf{x}_{\text{ast}}$ that captures the overall syntactic structure and variable dependencies of the smart contract.

In addition to extracting AST features, we also extract features from the Control Flow Graph (CFG). The CFG captures the execution flow within the program, showing the control dependencies between different code blocks, particularly in function calls, conditional branches, loops, and other control structures. Each node in the CFG represents a basic block composed of multiple EVM instructions, with edges indicating the control flow paths between these basic blocks, summarizing the program's execution paths. The CFG is represented as a directed graph $G = (V, E)$, where $V$ represents the nodes corresponding to basic blocks, each containing a set of sequential EVM instructions, and $E$ represents the edges between nodes. The CFG construction process is shown in the Appendix A.2.

To extract features for each basic block, we use a TextCNNZhang et al. (2023) model to process the sequence of EVM instructions embedded using Word2Vec. Each basic block consists of a sequence of EVM instructions, represented as $B_i = [I_{i1}, I_{i2}, \dots, I_{iN}] \in \mathbb{R}^{N \times d}$, where $N$ is the number of EVM instructions, and $d$ is the embedding dimension. These instructions are embedded using a pre-trained Word2Vec model that maps each instruction to a fixed-length vector. The TextCNN model

applies K convolutional filters, each with $R$ convolutional kernels of varying sizes $h \times d$ to capture features at different scales. The output feature vector obtained by the $k$-th filter is shown as follows.

$$f_i^{(k)} = \text{ReLU}(W_k * B_i + b_k) \tag{3}$$

where $W_k \in \mathbb{R}^{h \times d}$ is the weight matrix of the $k$-th convolutional filter, $b_k$ is the bias, and $f_i^{(k)}$ is the feature vector after applying the activation function ReLU. Each filter slides over the input sequence to aggregate local structural features of varying lengths. After obtaining the feature maps, a max-pooling operation is performed to extract the most significant feature from each feature map: $h_{\text{block}} = \max(f_i^{(1)}, f_i^{(2)}, \ldots, f_i^{(M)})$ where $h_{\text{block}}$ is the global feature vector for the basic block. The feature vectors for all basic blocks are concatenated to form the complete feature matrix $C$. The loss function used is the cross-entropy loss, defined as:

$$\mathcal{L}_{\text{CE}} = - \sum_{i=1}^{N} y_i \log(\hat{y}_i) \tag{4}$$

where $y_i$ is the true label, and $\hat{y}_i$ is the predicted probability for the $i - th$ sample, which ensures that TextCNN can capture multi-scale local structural features from the EVM instruction sequences.

After extracting these basic block features, the Graph Convolutional Network (GCN) Fan et al. (2021) processes the entire CFG using these features as node-level inputs to generate a global CFG representation. The feature vector of each node (basic block) $h_{\text{block}}$ is used as input to the GCN, represented as $H = \{h_1, h_2, \ldots, h_n\}$. Each edge $e_k$ in the CFG is embedded as a vector $e_k$, which encodes the relationship between nodes, such as function calls, conditional branches, and loops. To enable the transmission of information between nodes, the feature vector of the start node $h_{\text{start}_k}$ is concatenated with the edge embedding $e_k$ to form the input message vector $x_k = [h_{\text{start}_k} || e_k]$ where $||$ denotes the concatenation operation. This step fuses the node features and edge information, allowing the message passing process to capture contextual control flow information. Next, the GCN computes the message $m_k$ transmitted along each edge using a message generation network.

$$m_k = W_m' \cdot x_k + b_m' \tag{5}$$

where $W_m'$ is a learnable weight matrix, and $b_m'$ is a bias term. The message $m_k$ in Equation 5 represents the information passed from the start node to the end node of the edge.

To update the feature of each node $h_v^{(l+1)}$, the aggregated messages $\sum_{u \in \mathcal{N}(v)} m_u$ from its neighboring nodes are combined with the node's features $h_v^{(l)}$ using a non-linear transformation as follows.

$$h_v^{(l+1)} = \phi \left( W_v' \cdot \sum_{u \in \mathcal{N}(v)} m_u + U_v' \cdot h_v^{(l)} + b_v' \right) \tag{6}$$

where $\phi$ is the ReLU activation function, and $W_v'$, $U_v'$, and $b_v'$ are learnable parameters. Integration of information from neighboring nodes and the updating of the node's state. During the node update process, to distinguish the impact of different neighboring nodes, the GCN introduces a multi-head attention that assigns varying weights to neighboring nodes as shown in Equation 7.

$$\beta_{vu} = \frac{\exp\left(\text{LeakyReLU}(q^T[Ph_v' || Ph_u'])\right)}{\sum_{w \in \mathcal{N}(v)} \exp\left(\text{LeakyReLU}(q^T[Ph_v' || Ph_w'])\right)} \tag{7}$$

where $\beta_{vu}$ represents the attention weight assigned to the neighbor node $u$ for the target node $v$, and $q$ and $P$ are learnable parameters. Once all node features are updated, the GCN applies a weighted pooling operation to generate a global feature representation of the CFG as shown in Equation 8.

$$X_{\text{CFG}} = \sum_{v \in V} \beta_v h_v^{(L)} \tag{8}$$

where $\beta_v$ is the attention weight for node $v$, indicating its contribution to the global graph embedding. This pooling operation adaptively aggregates the node-level features, producing a comprehensive global embedding of the CFG that effectively captures control flow semantics.

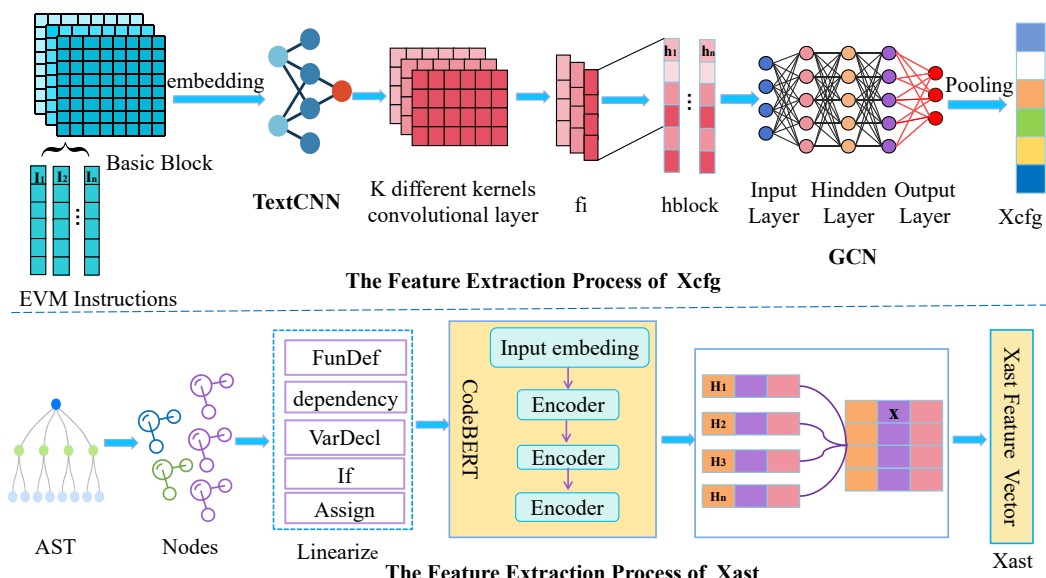

Figure 2: The Feature Extraction Process of $X_{ast}$ & $X_{cfg}$

To ensure compatibility during feature fusion, a linear transformation is applied to the global feature vectors of both the AST and CFG. Specifically, the AST feature vector $X_{AST}$ and the CFG feature vector $X_{CFG}$ are mapped to a common dimensional space. $W_{AST} \in \mathbb{R}^{d' \times d_{AST}}$ and $W_{CFG} \in \mathbb{R}^{d' \times d_{CFG}}$ are learnable weight matrices, and $b_{AST}, b_{CFG}$ are bias terms. This transformation aligns the dimensions of the AST and CFG representations, ensuring that they are compatible with subsequent combinations. After the linear transformation, the normalized feature vectors are fused to form a comprehensive feature representation:

$$X_{combined} = \alpha \cdot X'_{AST} + (1 - \alpha) \cdot X'_{CFG} \tag{9}$$

where $\alpha$ is a learnable parameter that controls the relative contribution of the AST and CFG features. The model is trained to optimize the feature fusion weights with the loss function in Equation 10

$$\min_{W_{cfg}, W_{ast}} \sum_{i=1}^{N} \mathcal{L}\left( f(\mathbf{x}^{(i)}_{combined}), y^{(i)} \right) \tag{10}$$

where $\mathcal{L}$ denotes the loss function (e.g., cross-entropy), $f(\mathbf{x}^{(i)}_{combined})$ represents the model's predicted output for the $i$-th sample, and $y^{(i)}$ is the corresponding true label.

After obtaining the optimized combined feature vector $\mathbf{x}_{combined}$, the next step involves integrating it with known defect information. Known defect code snippets are converted into high-dimensional vector representations using a pre-trained text embedding model (text-embedding-004). This model processes each defect code snippet and outputs an embedding vector $\mathbf{d}_{embedding}$, that encodes semantic information related to the defect type. To address the issue of increased feature dimensionality when concatenating $\mathbf{x}_{combined}$ and $\mathbf{d}_{embedding}$, we introduce a dimension alignment module before the concatenation step to project both feature vectors into a unified low-dimensional space.

$$\mathbf{x}'_{combined} = W^{(combined)}_{low} \cdot \mathbf{x}_{combined} + b^{(combined)}_{low} \tag{11}$$

$$\mathbf{d}'_{embedding} = W^{(embedding)}_{low} \cdot \mathbf{d}_{embedding} + b^{(embedding)}_{low} \tag{12}$$

where $W^{(combined)}_{low}$ and $W^{(embedding)}_{low}$ are learnable projection matrices, and $b^{(combined)}_{low}, b^{(embedding)}_{low}$ are bias terms. The attention-weighted combination is performed as follows.

$$\mathbf{E} = \alpha \cdot \mathbf{x}'_{combined} + (1 - \alpha) \cdot \mathbf{d}'_{embedding} \tag{13}$$

where $\alpha$ is a parameter in the structural features $\mathbf{x}'_{\text{combined}}$ and the semantic features $\mathbf{d}'_{\text{embedding}}$.

The resulting matrix $\mathbf{E} = [\mathbf{E}_1, \mathbf{E}_2, \ldots, \mathbf{E}_k]$ is constructed, where $\mathbf{E}_k$ represents the combined feature embedding for the $k$-th known defect type. Each defect type has its own corresponding defect embedding, ensuring that the representation in $E$ is comprehensive and covers various vulnerabilities. These combined matrices, including matrix $E$, are stored in the vector database (VecDB). Table 1 describes the definitions and corresponding scenarios for 4 common NFT smart contract defects.

Table 1: The definitions for NFT smart contract defects

| Defect Type | Definition | Scenario Description |
| --- | --- | --- |
| ERC-721 Reentrancy | Modifies the state variable after the external invocation.Yang et al. (2023) | A malicious onERC721Received function can reenter the victim contract during a token transfer, disrupting its logic. This may allow the minting of more NFTs than intended, causing economic loss. |
| Public Burn | Does not check the caller of the burn operation on NFT.Yang et al. (2023) | If the burn function is not restricted to the NFT owner, anyone could burn another's NFT without permission. This could lead to the unintended destruction of all NFTs in a project. |
| Risky Mutable Proxy | Makes the proxy contract modifiable.Yang et al. (2023) | If the proxy registry address is modifiable, an attacker could change it and transfer all tokens without permission. This poses a significant risk as the proxy can act on behalf of the user. |
| Unlimited Minting | When there is no check on the max supply during the minting process.Yang et al. (2023) | Without verifying whether the current supply exceeds the promised limit, the contract owner or an attacker could exploit a reserve function to mint unlimited NFTs. |

## 2.2 KNOWLEDGE BASE RETRIEVAL

During the knowledge base retrieval process, the system compares the generated vectors with the known defect vectors stored in the VectorDB. To evaluate the similarity between the vectors, the tool combines the use of cosine similarity and Euclidean distance. Cosine similarity effectively measures the similarity between vectors by calculating the angle between them, with values closer to 1 indicating higher similarity. Euclidean distance, on the other hand, considers the absolute magnitude of the vectors, capturing quantitative differences in contract operations such as the number of minted tokens or function call counts. The following is the mathematical formula for cosine similarity. Cosine similarity ($\cos \theta$) measures the similarity between two vectors by calculating the dot product of the vectors divided by the product of their magnitudes. The formula is given as follows:

$$\cos \theta = \frac{\mathbf{A} \cdot \mathbf{B}}{\|\mathbf{A}\|\|\mathbf{B}\|} = \frac{\sum_{i=1}^{n} A_i B_i}{\sqrt{\sum_{i=1}^{n} A_i^2} \sqrt{\sum_{i=1}^{n} B_i^2}} \tag{14}$$

where $\mathbf{A}$ and $\mathbf{B}$ represent the contract vector under analysis and the known defect vector, $A_i$ and $B_i$ are the $i$-th components of vectors $\mathbf{A}$ and $\mathbf{B}$, respectively, $n$ is the dimension of the vectors. To capture the absolute magnitude of feature values, we use Euclidean distance calculated as follows.

$$d(A, B) = \sqrt{\sum_{i=1}^{n} (A_i - B_i)^2} \tag{15}$$

The Equation 15 measures the absolute distance between two vectors in feature space, allowing us to consider the structural similarity of the contracts and their differences in quantity and scale. As illustrated in Figure 3, the process begins by selecting defect types with high similarity from the knowledge base retrieval phase, where cosine similarity is calculated to measure the alignment between the contract vectors and known defect vectors. Euclidean distance is employed to capture the magnitude differences between the contract and defect vectors.

## 2.3 LLM REASONING

In the LLM reasoning phase, NATLM performs the final stage of defect detection and analysis using an LLM. Once the most relevant defect types are identified through the combination of cosine

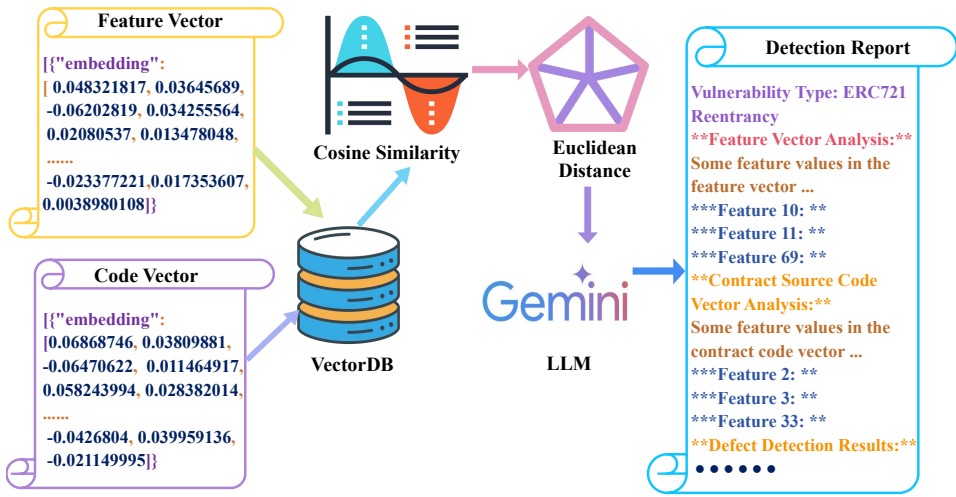

Figure 3: LLM Reasoning Process Detection Report

similarity and Euclidean distance, these feature vectors, along with the contract's feature and code vectors, are input into the LLM for in-depth analysis. The LLM further interprets and validates the similarity, deeply analyzing the contextual relationships between the identified defect type and the current contract. The similarity-based retrieval results are obtained by adjusting the model's sensitivity to imbalanced data distributions through a weighted loss function. The weight assigned to each defect type is calculated as the Equation 16.

$$w_i = \frac{1}{\log(1 + n_i)} \tag{16}$$

where $n_i$ represents the sample count for the $i$-th defect type. This weighting scheme ensures that defect types with fewer samples receive higher weights, thereby increasing their influence during gradient updates and improving the model's performance on rare defect types. During the training process, the weighted loss function is defined as 17.

$$\mathcal{L}_{\text{weighted}} = -\sum_{i=1}^{N} w_i y_i \log(\hat{y}_i) \tag{17}$$

where $N$ is the total number of samples, $y_i$ and $\hat{y}_i$ represent the true label and predicted probability for the $i$-th sample, respectively, and $w_i$ is the corresponding weight for the defect type. The feature vectors of these identified defects, along with the feature vectors and code vectors of the current contract, are then input into the LLM for in-depth reasoning. To improve the accuracy of defect detection results and reduce false positives, a confidence threshold filtering strategy is introduced during the output stage. To prevent low-confidence predictions from affecting the analysis report, an adjustable confidence threshold $\tau$ is set. When the confidence score $\hat{p}_i$ of a defect type exceeds $\tau$, the result is retained and recorded in the analysis report. Conversely, predictions with confidence scores below $\tau$ are filtered out to prevent unreliable results, following the rule in the Equation 18.

$$\hat{y}_i = \begin{cases} y_i, & \text{if } \hat{p}_i \geq \tau, \\ \varnothing, & \text{if } \hat{p}_i < \tau, \end{cases} \tag{18}$$

where $\hat{p}_i$ denotes the confidence score for the $i$-th defect type, and $y_i$ represents the corresponding predicted label. After filtering low-confidence predictions, the LLM generates a comprehensive detection report. This report includes a detailed analysis of the identified defect types, their potential security impacts, and recommended remediation measures.

## 3 EXPERIMENT

To construct a dataset of NFT smart contracts, we utilized the publicly available Smart Contract Sanctuary on GitHub Ortner & Eskandari (2024), which aggregates verified smart contracts from

multiple blockchains (e.g., Ethereum, BSC, Polygon, etc.). To extract relevant NFT smart contracts, we apply a pattern-matching approach based on ERC-721 interface definitions and related keywords. Specifically, we search for interface function signatures, such as supportsInterface(bytes4 interfaceID) and ownerOf(uint256 tokenId), alongside keywords such as "NFT" and "ERC721" in contract names, comments, and function signatures. We compared extracted contracts against interface definitions to confirm adherence to relevant standards. We incorporate data from Yang et al. (2023) to the dataset. To avoid redundancy when merging the two datasets, we compare the addresses of the contracts and remove duplicates. After applying consistent filtering and deduplication criteria, we obtain a dataset containing 8,672 NFT smart contracts.

In the experimental setup, CodeBERT adopts a two-layer MLP and is trained for 30 epochs with a batch size of 128, a learning rate of 0.0003, a dropout rate of 0.1, and the AdamW optimizer. TextCNN is with a batch size of 256, a learning rate of 0.0003, and the Adam optimizer. The GCN contains two convolutional layers, each with a hidden dimension of 128. The GCN is trained using the Adam optimizer, with a batch size of 64, a learning rate of 0.001, and a dropout rate of 0.5. We set the Temperature $T$ as 0.7, the reason is shown in Appendix A.3.

To compare the detection performance of NATLM with other state-of-the-art (SOTA) smart contract vulnerability detection tools, we selected 10 tools, including symbolic execution-based tools (Mythril, OyenteLuu et al. (2016)), static analysis tools ( SecurifyTsankov et al. (2018), Slither, SailfishBose et al. (2022), ACheckerGhaleb et al. (2023)), and machine learning-enhanced LLMs for smart contract analysis. The comparison also evaluates the detection capabilities of LLMs models (GPT-3.5-turbo, GPT-4, GPT-4o, and Gemini pro-1.5) for NFT-specific vulnerabilities. Some tools are constrained to predefined vulnerability categories and cannot generalize to detect vulnerabilities beyond their predefined rules. In contrast, LLM tools demonstrate the potential to adapt to new vulnerability categories. Table 2 illustrates the performance of NATLM and 10 baseline detection tools across six smart contract vulnerability categories (i.e., Reentrancy, Access Control, ERC-721 Reentrancy, Public Burn, Risky Mutable Proxy, and Unlimited Minting). The results indicate that while traditional symbolic execution and static analysis tools perform well for conventional vulnerabilities (such as reentrancy), they struggle to detect NFT-specific vulnerabilities (e.g., ERC-721 reentrancy and unlimited minting). In contrast, LLMs demonstrate greater adaptability across different categories. However, they still suffer from moderate rates of false positives and negatives. NATLM outperforms traditional tools and LLM baselines by consistently achieving high precision and recall across all types of vulnerabilities.

Table 2: Comparison of Detection Capabilities of NATLM and Baseline Tools Smart Contract Vulnerabilities. ✓ indicates always identifies **O** indicates partial identification, and ✗ indicates low precision or complete failure to detect this vulnerability.

| Tool | Reentrancy | Access Control | ERC-721 Reentrancy | Public Burn | Risky Mutable Proxy | Unlimited Minting |
|------|-----------|----------------|--------------------|-------------|--------------------|--------------------|
| Mythril | O | O | ✗ | ✗ | ✗ | ✗ |
| Securify | ✓ | ✗ | ✗ | ✗ | ✗ | ✗ |
| Oyente | ✓ | ✗ | ✗ | ✗ | ✗ | ✗ |
| Slither | ✓ | ✗ | ✗ | ✗ | ✗ | ✗ |
| Sailfish | ✓ | ✗ | ✗ | ✗ | ✗ | ✗ |
| AChecker | ✗ | ✓ | ✗ | ✗ | ✗ | ✗ |
| GPT-3.5-turbo | O | ✗ | ✗ | ✗ | ✗ | ✗ |
| GPT-4 | O | O | O | O | O | O |
| GPT-4o | O | O | O | O | O | O |
| Gemini pro-1.5 | O | O | O | O | O | O |
| **NATLM** | ✓ | ✓ | ✓ | ✓ | ✓ | ✓ |

In the experiment, we categorize the detection results of NATLM into three classes: True Positive (TP), False Positive (FP), and False Negative (FN). The results, there are 503 NFT smart contracts with ERC-721 Reentrancy defects, 44 with Public Burn defects, 15 with Risky Mutable Proxy defects, and 781 with Unlimited Minting defects in the dataset. NATLM detects 482 ERC-721 Reentrancy defects, of which 423 are correctly classified as true positives (TP), 59 as false positives (FP), and 80 as false negatives (FN). For the Public Burn defect, NATLM detects 49 contracts with the defect, with 42 correctly classified as TP, 7 as FP, and 2 as FN. For the Risky Mutable Proxy defect, NATLM detects 14 contracts with the defect, with 13 correctly classified as TP, 1 as FP, and 2 as FN. Finally, for the Unlimited Minting defect, NATLM detects 836 contracts with the defect, with 712 correctly classified as TP, 124 as FP, and 69 as FN.

Table 3 presents the comparative performance of NATLM and four LLM baselines(GPT-3.5-turbo, GPT-4, GPT-4o, and Gemini pro-1.5) across five categories of smart contract vulnerabilities: Reen-

trancy, ERC-721 Reentrancy, Public Burn, Risky Mutable Proxy, and Unlimited Minting. The evaluation is conducted on a dataset comprising 8,672 NFT smart contracts.

Table 3: The performance metrics of NATLM are compared against four LLM models, evaluated based on Precision (PRE), Recall (RE), and F1-score (F1).

| Tool | Reentrancy | | | ERC-721 Reentrancy | | | Public Burn | | | Risky Mutable Proxy | | | Unlimited Minting | | |
|---|---|---|---|---|---|---|---|---|---|---|---|---|---|---|---|
| | PRE | RE | F1 | PRE | RE | F1 | PRE | RE | F1 | PRE | RE | F1 | PRE | RE | F1 |
| GPT-3.5-turbo | 26.03% | 38% | 30.89% | 23.5% | 41.96% | 30.12% | 19.16% | 35% | 24.76% | 15.2% | 28.51% | 19.83% | 17.5% | 32.61% | 22.78% |
| GPT-4 | 33.5% | 87.1% | 48.39% | 32.02% | 82.8% | 46.18% | 25.12% | 80.2% | 38.26 % | 28.91% | 78.75% | 42.29% | 30.51% | 83.55% | 44.7% |
| GPT-4o | 30.17% | 83.92% | 44.38% | 28.2% | 79.61% | 41.65% | 22.5% | 77.43% | 34.86% | 30.5% | 74.8% | 43.32% | 33.19% | 80.92% | 47.07% |
| Gemini pro-1.5 | 31.76% | 80.2% | 45.5% | 31.62% | 81.5% | 45.56% | 32.18% | 76.5% | 45.3% | 32.89% | 75.08% | 45.74% | 34.17% | 82.52% | 48.33% |
| **NATLM** | **87.12%** | **90.58%** | **88.82%** | **87.75%** | **84.09%** | **85.88%** | **85.74%** | **95.45%** | **90.32%** | **92.85%** | **86.66%** | **89.65%** | **85.16%** | **91.16%** | **88.06%** |

For Reentrancy, GPT-3.5-turbo performs poorly, with an F1-score of 30.89%, precision of 26.03%, and recall of 38%. This suggests that GPT-3.5-turbo may struggle to distinguish between vulnerable and non-vulnerable contracts, potentially due to limitations in its contextual understanding or token length constraints. In contrast, GPT-4 shows a substantial improvement in recall (87.1%), but its precision remains low at 33.5%. This indicates that while GPT-4 can detect the most vulnerable contracts, it generates many false positives. For ERC-721 Reentrancy, NATLM achieves an F1-score of 85.88%, significantly outperforming GPT-4 (46.18%) and Gemini pro-1.5 (45.56%). The relatively low precision scores of the LLM baselines (ranging from 23.5% to 32.02%). NATLM achieves an F1-score of 90.32% for Public Burn vulnerabilities, indicating a highly accurate detection of unauthorized token destruction. In comparison, GPT-4 achieves a recall of 80.2% but a much lower precision (25.12%), resulting in an F1-score of 38.25%. This suggests that GPT-4 can detect many true vulnerabilities, with a high false positive rate. For Risky Mutable Proxy and Unlimited Minting, NATLM maintains F1-scores of 89.65% and 88.06%. In contrast, the LLM baselines achieve F1-scores ranging from 32% to 48%, with GPT-4o demonstrating slightly better recall but consistently low precision. Overall, NATLM outperforms the baseline tools in detecting vulnerabilities in NFT smart contracts, achieving an overall Precision of 87.72%.

# 4 RELATED WORK

In the smart contract vulnerability detection field, numerous detection tools have emerged. The static analysis tool SlitherFeist et al. (2019) detects potential vulnerabilities and coding errors by inspecting the smart contracts' bytecode and source code. Tools like Manticor e Mossberg et al. (2019) and MythrilSharma & Sharma (2022) utilize symbolic execution to analyze smart contract behavior by exploring possible execution paths and identifying potential vulnerabilities and anomalies. Additionally, Liu et al. (2021) combines graph neural networks with expert knowledge for smart contract vulnerability detection by transforming the control and data flow of the source code into a contract graph, normalizing it, and using temporal message propagation.

GPTLENS Hu et al. (2023) introduces an adversarial framework that divides the traditional single-stage detection process into two collaborative phases: generation and discrimination. In this framework, the LLM is designed to play two adversarial roles: the auditor and the critic. LLM4Fuzz Shou et al. (2024) integrates fuzz testing with LLMs to guide and organize fuzz testing activities. Liu et al. Liu et al. (2024) propose PropertyGPT, a system that leverages large language models like GPT-4 to generate formal verification properties for smart contracts automatically. The system addresses the challenges of developing properties that are compilable, appropriate, and runtime-verifiable by utilizing a vector database for property retrieval, iterative revision with external feedback, and formal verification through a dedicated prover.

# 5 CONCLUSION

In this study, we propose the NATLM framework, which combines static analysis and LLM (Gemini Pro 1.5) to detect defects in NFT smart contracts. The strength of the NATLM framework lies in its combination of structured code analysis from static analysis with the deep semantic understanding capabilities of the LLM. We conducted experiments on a dataset of 8,672 NFT smart contracts, achieving an overall precision of 87.72%.

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

## A  APPENDIX

### A.1  THE USE OF LARGE LANGUAGE MODELS (LLMS)

During the paper writing process, LLMs have been utilized for English translation and polishing.

### A.2  THE CFG CONSTRUCTION PROCESS

The Control Flow Graph (CFG) serves as a representation of the execution flow within a program, illustrating the control dependencies among various code blocks. This representation is especially pertinent in the context of function calls, conditional branches, loops, and other control structures.

Figure 4 illustrates the process of constructing the CFG. In the CFG, each node corresponds to a basic block, which comprises multiple instructions from the Ethereum Virtual Machine (EVM). The edges between these nodes denote the potential control flow paths, thereby providing a comprehensive overview of the program's execution paths.

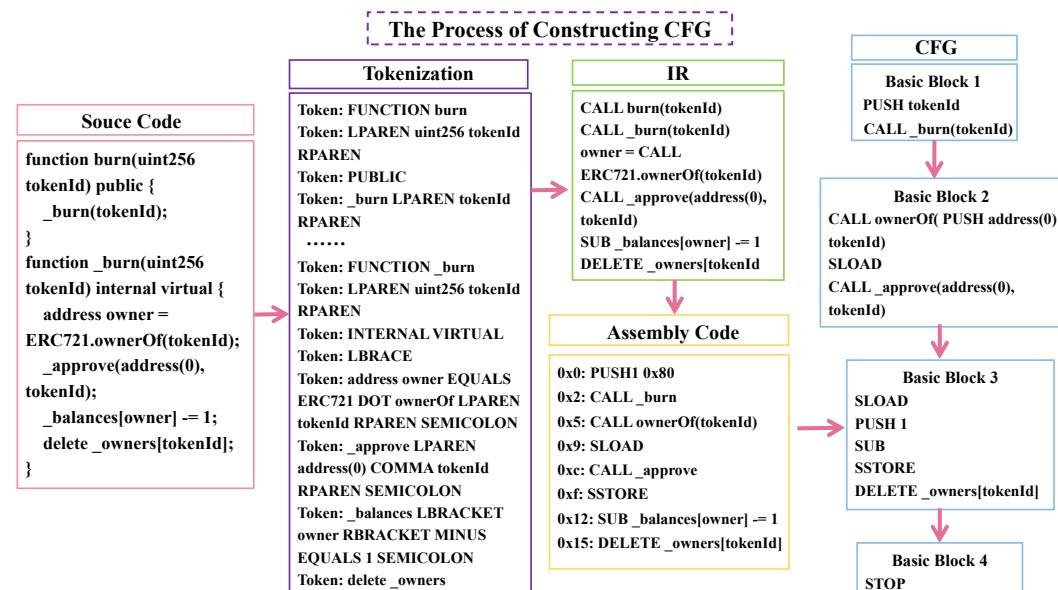

Figure 4: The Process of Constructing CFG

## A.3 THE TEMPERATURE SETTING

Temperature is an important hyperparameter that controls the randomness of model-generated results. When the temperature is high, the diversity of the generated results increases, but this may come at the cost of accuracy. Conversely, when the temperature is low, the generated results tend to be consistent with reduced randomness.

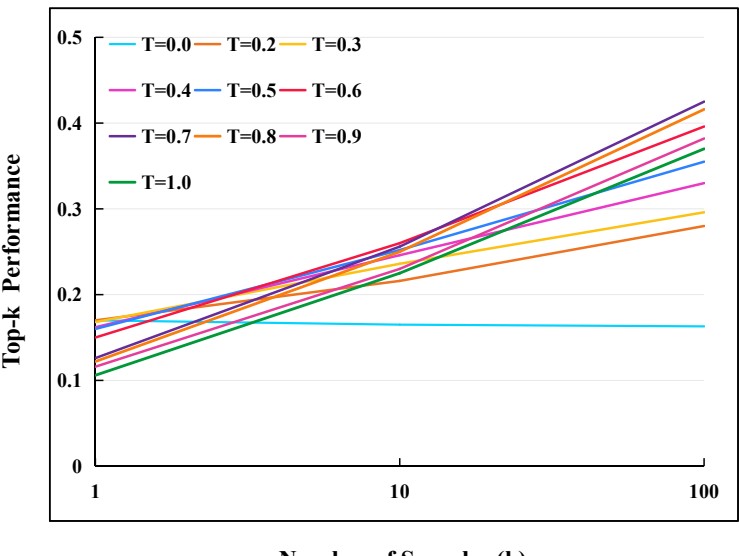

Figure 5: Effect of Temperature on Top-K Performance Across Samples

As shown in Figure 5, the chart presents the model's performance under different temperature (T) settings while generating different samples. As the number of generated samples (k) increases, a higher temperature setting (T=0.7) achieves the best Top-k Performance, indicating that in scenarios with a large number of samples, a higher temperature helps improve the quality of the generated audit reports.

