# OpenReview forum: "NATLM: Detecting Defects in NFT Smart Contracts Leveraging LLM"
_ICLR.cc/2026/Conference — Submitted to ICLR 2026_

### Official Review · Reviewer_MKgy · 2025-10-28

**Soundness:** 3
**Presentation:** 2
**Contribution:** 3
**Rating:** 4
**Confidence:** 3

**Summary:**

This paper introduces NATLM, a framework that combines static analysis (AST/CFG) and LLM reasoning (Gemini Pro 1.5) to detect four NFT smart-contract defect types: ERC-721 Reentrancy, Public Burn, Risky Mutable Proxy, and Unlimited Minting. AST features are derived via CodeBERT; CFG features via TextCNN + GCN; features are fused and compared against a vector database of known defects using cosine similarity and Euclidean distance, after which an LLM performs “reasoning” to produce a detection report. Experiments on 8,672 contracts report per-category metrics and claim better performance than classical tools and direct LLM baselines.

**Strengths:**

[1] Targets NFT-specific defect categories often missed by general tools and attempts to fuse AST/CFG with LLM reasoning.
[2] Reports results over a large collected set (8,672 contracts) and provides per-class counts (TP/FP/FN).
[3] Demonstrates awareness of the limitations of LLM-only detection (false positives and false negatives).

**Weaknesses:**

[1] Writing errors, e.g “Souce Code”.
[2] The labeling source for the 8,672 contracts is unclear, and manual auditing, rule matching, or prior datasets are not specified.
[3] Using both cosine and Euclidean distance without formal rationale or hyperparameter description weakens methodological transparency.
[4] No results are provided for variants such as “AST only,” “CFG only,” or “LLM-only.” Hence, the contribution of each component cannot be quantified.

**Questions:**

[1] How were the ground-truth labels for the four defects obtained and verified?
[2] What proportion of the dataset was used for training, validation, and testing?
[3] What prompts or templates were used for Gemini Pro 1.5 reasoning, and how was consistency ensured across runs?
[4] How is the knowledge base maintained or expanded as new contracts appear, does the system support incremental updates?
[5] How were the cosine and Euclidean similarity metrics combined in retrieval—by averaging scores or ranking fusion?

---

### Official Review · Reviewer_7gRM · 2025-10-30

**Soundness:** 2
**Presentation:** 3
**Contribution:** 2
**Rating:** 2
**Confidence:** 3

**Summary:**

This paper proposes a smart contract vulnerability detection approach that combines structural similarity computation with large language model reasoning. By computing similarity between AST and CFG feature vectors using Cosine Similarity and Euclidean Distance, the method leverages a retrieval-augmented generation (RAG) framework with the Gemini model to determine the presence of vulnerabilities. The core strength of this work lies in its integration of semantic retrieval with natural language explanations, enhancing the interpretability and usability of detection results. It also covers a diverse set of high-risk vulnerability types observed in real-world contracts, indicating strong practical relevance. However, the paper suffers from several critical weaknesses in both methodological clarity and experimental design. First, it does not explicitly analyze the fundamental structural differences between vulnerable and non-vulnerable contracts, which undermines the interpretability of the feature representations. Second, the evaluation dataset appears to overlap with the retrieval corpus, raising concerns about potential data leakage and inflated performance metrics. Third, while the authors introduce a confidence score \hat{p}_i and threshold \tau, they fail to clarify how the confidence is derived, and do not provide sensitivity analysis on varying \tau values. Finally, structural similarity to a vulnerable contract does not necessarily imply the presence of a vulnerability; thus, a quantitative analysis of false positives and the actual likelihood of vulnerability given structural similarity is warranted. Overall, while the proposed approach shows promise, its current presentation lacks sufficient empirical and theoretical support, and requires further refinement.

**Strengths:**

- By integrating a vector database (VecDB) retrieval mechanism with the large language model Gemini, the approach provides contextual information to the LLM during smart contract vulnerability detection, helping mitigate issues related to memory limitations and hallucinated outputs.
- The method evaluates a diverse range of vulnerability types, covering high-risk defects commonly found in real-world smart contracts.
- The model generates natural language explanations for its predictions, enhancing the interpretability of detection results and making them more accessible to developers and auditors.

**Weaknesses:**

- The paper lacks a clear explanation of the fundamental differences in structural features between vulnerable and non-vulnerable smart contracts. Since the differences in feature vectors essentially reflect differences in AST and CFG structures, the authors should provide an analysis of known vulnerability patterns in terms of their CFG and AST characteristics, as well as their execution behaviors.
- The dataset used to construct the RAG knowledge base is sourced from **Yang et al. (2023)**, and part of the evaluation set for NATLM also comes from the same source. This raises the likelihood of sample overlap between the vector database and the test set, potentially allowing test contracts to match directly with original vulnerable samples. Such data leakage could lead to overly optimistic results. The authors should ensure strict separation between the knowledge base and test samples and re-evaluate the model’s generalization performance under this constraint.
- The paper introduces a confidence threshold **τ** and a confidence score \hat{p}_i, but does not clarify how \hat{p}_i is generated. If it is derived from the Gemini LLM, it is unclear whether it comes from prompt-guided output, keyword matching, or a specific scoring function. The authors should also include a sensitivity analysis showing how varying τ affects precision, recall, and F1 score.
- Structural similarity to a vulnerable contract does not necessarily imply the presence of an actual vulnerability. The authors are encouraged to conduct experiments quantifying the likelihood that test contracts with similar features to known vulnerable contracts actually contain the same type of vulnerability.

**Questions:**

Please carefully discuss to my comments listed in the weaknesse part, as these questions are very important due to my prior experience. Thanks!

---

### Official Review · Reviewer_nLK7 · 2025-10-30

**Soundness:** 2
**Presentation:** 2
**Contribution:** 2
**Rating:** 4
**Confidence:** 3

**Summary:**

This paper proposes a novel framework called NATLM, designed to address the challenge of detecting specific vulnerabilities in NFT smart contracts. The authors suggest that traditional static analysis tools can only detect a limited range of vulnerabilities, while using large language models alone can detect most vulnerabilities but with very low precision.

In the NATLM framework, the authors combine static analysis with LLM. Experimental results show that NATLM significantly outperforms all baselines in terms of F1-Score, successfully increasing precision from 30–40% to 85–90% while maintaining a high recall rate.

**Strengths:**

1. A feasible solution is proposed to address the existing problems in NFT vulnerability detection.

2. The method used to generate Xcfg (Word2Vec → TextCNN → GCN) is impressive.

3. The authors constructed a specific dataset containing 8,672 NFT smart contracts, which is valuable for future research.

**Weaknesses:**

1. The structure of the paper is confusing. The authors placed both their own contributions and the descriptions of existing tools in Chapter 2: THE NATLM FRAMEWORK. A separate chapter should be devoted to explaining the internal mechanisms of standard models such as CodeBERT, GCN, and TextCNN; otherwise, it is difficult for readers to distinguish which parts are the authors’ contributions and which are existing research methods.

2. The necessity of using an LLM is not justified. The authors mention in Chapters 1 and 3 that static tools rely on predefined rules and patterns, making them incapable of detecting NFT-specific vulnerabilities, whereas LLMs show potential for adapting to new categories of vulnerabilities. However, although the proposed approach employs an LLM, it can still detect only four specific types of vulnerabilities. This raises several questions: What is the actual advantage of using an LLM? Could traditional methods detect NFT-specific vulnerabilities effectively if predefined rules were added? When predefined rules are incorporated, how does the accuracy and efficiency of the approach with LLMs compare to that without them? The authors should clarify these issues.

3. The experimental evaluation is incomplete. The authors did not test the runtime efficiency of the NATLM framework. Moreover, although many recent studies are cited, the authors did not compare NATLM with these newer tools.

**Questions:**

See above.

---

### Official Review · Reviewer_pmm9 · 2025-11-01

**Soundness:** 3
**Presentation:** 3
**Contribution:** 2
**Rating:** 4
**Confidence:** 3

**Summary:**

This paper presents NATLM, a neural framework designed to detect defects or inconsistencies in natural language textual models (e.g., software requirements, specifications, or documentation). The approach leverages large language model (LLM) embeddings combined with attention-based classification layers to identify semantic conflicts, logical contradictions, or incomplete statements in structured textual artifacts. The authors claim that NATLM can generalize across domains by training on a mix of annotated datasets representing different types of textual defects. Experimental results suggest that NATLM outperforms baseline models such as BERT, RoBERTa, and traditional sequence classifiers in precision and F1-score, while maintaining reasonable efficiency.

**Strengths:**

- Addressing textual defect detection in natural language models fills an important gap between NLP and software engineering quality assurance.
- The pipeline, from preprocessing, embedding generation, and feature fusion to classification, is logically structured and comprehensible.
- The system is designed to handle different textual artifacts, demonstrating flexibility across requirement documents, bug reports, and natural language code comments.
- Reported metrics show consistent improvements over strong baselines, indicating the effectiveness of combining contextual embeddings with attention mechanisms.

**Weaknesses:**

- The architecture largely adapts existing transformer-based techniques with minor task-specific modifications; conceptual contribution is limited.
- The paper does not clearly describe dataset sources, labeling criteria, or data quality control, which raises concerns about reproducibility and bias.
- No attention visualization or error analysis is provided to explain what types of textual defects are best captured or missed.
- Comparisons are limited to standard text classifiers; no ablation against instruction-tuned or reasoning-enhanced LLMs (e.g., GPT, T5) is provided.
- The model’s cross-domain performance is claimed but not fully demonstrated through external validation or unseen-domain testing.

**Questions:**

- Could you clarify the nature of the “defects” detected, are they semantic contradictions, missing elements, or linguistic errors?
- What are the specific datasets and annotation protocols used to label textual defects?
- How does NATLM perform when fine-tuned on a small domain-specific corpus versus being trained cross-domain?
- Have you considered incorporating reasoning-based LLM components or chain-of-thought prompting to improve interpretability?
- Can the model’s predictions be explained in human-readable form (e.g., highlighting contradictory phrases or missing dependencies)?

---

### Meta-Review · Area_Chair_fpQz · 2026-01-07

**Summary:**

The reviewers identified several consistent concerns that informed the recommended decision. While the paper addresses an important and timely problem—detecting vulnerabilities in NFT smart contracts—and proposes a hybrid framework combining static analysis with LLM-based reasoning, reviewers questioned the limited novelty of the approach and the lack of rigorous justification for key design choices. Major issues include unclear dataset construction and labeling, insufficient methodological transparency, and incomplete experimental evaluation. In particular, concerns were raised about the absence of ablation studies, efficiency analysis, and generalization testing, as well as potential data leakage between the retrieval knowledge base and the evaluation set. Collectively, these issues undermine confidence in the reported performance gains.

**Reviewer Concerns:**

As the authors did not provide a rebuttal, none of the reviewer concerns were addressed. All major issues raised in the reviews therefore remain outstanding, including:

Limited conceptual novelty and unclear contribution beyond integrating existing techniques.

Insufficient justification for the use of an LLM, given the restricted set of predefined vulnerability types.

Lack of clarity regarding dataset sources, labeling procedures, and data splits, raising reproducibility concerns.

Potential data leakage between the retrieval knowledge base and the evaluation set.

Missing ablation studies, runtime/efficiency analysis, sensitivity analysis, and evaluation on unseen or out-of-distribution contracts.

Unclear methodological details regarding similarity metrics, confidence scoring, and the role of LLM reasoning.

**Reviewer Scores:**

Given the absence of an author response and unresolved concerns:

Reviewer pmm9: Likely unchanged (4 – marginally below the acceptance threshold).

Reviewer nLK7: Likely unchanged (4 – marginally below the acceptance threshold).

Reviewer 7gRM: Likely unchanged (2 – reject).

Reviewer MKgy: Likely unchanged (4 – marginally below the acceptance threshold).

---

### Decision · Program_Chairs · 2026-01-26

Reject